# Loss of Heterozygosity in the Circulating Tumor DNA and CD138+ Bone Marrow Cells in Multiple Myeloma

**DOI:** 10.3390/genes14020351

**Published:** 2023-01-29

**Authors:** Maiia Soloveva, Maksim Solovev, Elena Nikulina, Natalya Risinskaya, Bella Biderman, Igor Yakutik, Tatiana Obukhova, Larisa Mendeleeva

**Affiliations:** National Medical Research Center for Hematology, Novy Zykovski Lane 4a, 125167 Moscow, Russia

**Keywords:** multiple myeloma, plasmacytoma, STR-profile, loss of heterozygosity (LOH), plasma circulating tumor DNA (ctDNA), *RAS* gene family

## Abstract

Multiple myeloma (MM) is characterized by heterogeneity of tumor cells. The study of tumor cells from blood, bone marrow, plasmacytoma, etc., allows us to identify similarities and differences in tumor lesions of various anatomical localizations. The aim of this study was to compare the loss of heterozygosity (LOH) by tumor cells by assessing STR profiles of different MM lesions. We examined paired samples of plasma circulating tumor DNA (ctDNA) and CD138+ bone marrow cells in MM patients. For patients with plasmacytomas (66% of 38 patients included), the STR profile of plasmacytomas was also studied when biopsy samples were available. Diverse patterns of LOH were found in lesions of different localization for most patients. LOH in plasma ctDNA, bone marrow, and plasmacytoma samples was found for 55%, 71%, and 100% of patients, respectively. One could expect a greater variety of STR profiles in aberrant loci for patients with plasmacytomas. This hypothesis was not confirmed—no difference in the frequency of LOH in MM patients with or without plasmacytomas was found. This indicates the genetic diversity of tumor clones in MM, regardless of the presence of extramedullar lesions. Therefore, we conclude that risk stratification based on molecular tests performed solely on bone marrow samples may not be sufficient for all MM patients, including those without plasmacytomas. Due to genetic heterogeneity of MM tumor cells from various lesions, the high diagnostic value of liquid biopsy approaches becomes obvious.

## 1. Introduction

Multiple myeloma (MM) is a hematological malignancy originating from plasma cells. The wide variety of molecular genetic approaches allows the acquisition of arrays of data concerning the genetic structure of the tumor. MM molecular karyotype, gene expression profiles, gene mutations and their combinations, which adversely affect the course and prognosis of the disease, are being actively studied. MM is known to be a genetically unstable tumor. Primary and secondary genetic lesions can be indicated. The primary genetic lesions include chromosomal translocations involving the locus of the immunoglobulin heavy chain gene and hyperdiploidy with multiple copies of odd chromosomes. Secondary genetic events are aberrations such as deletions (deletion/monosomy of chromosome 13, deletion 1p, deletion 17p13), amplification of locus 1q21, translocations involving the locus of the *cMYC*/8q24 gene, and somatic gene mutations [1]. As monoclonal gammapathy of undetermined significance (MGUS) transforms into smoldering, symptomatic MM, somatic mutations, and new chromosomal aberrations accumulate. Plasma-cell leukemia, as well as extramedullary plasmacytoma in MM, are the most unfavorable forms of the disease, characterized by the multiple genetic lesions in tumor cells. In MM, the diverse factors of unfavorable prognosis form the concept of “high-risk myeloma”. These include somatic mutations in certain genes, certain cytogenetic aberrations, advanced disease according to the current ISS and R-ISS staging systems, phenotypic signs of high risk—the presence of plasmacytomas, refractory course of the disease, or early relapse. The median overall survival of patients from the high-risk myeloma is significantly lower than that of patients with the standard risk and is less than 2 years in patients that are not suitable for autologous hematopoietic cell transplantation (auto-HSCT), and less than 3 years in candidates for high-dose treatment [2]. The list of prognostic factors is being permanently updated based on the accumulation of new data. The current MM staging system relies on the genetic profiling of tumor plasma cells localized only in the bone marrow. Risk stratification based on cytogenetic testing of bone marrow samples in patients with plasmacytomas may lead to false negative results. Thus, no high-risk genetic aberrations may be detected in the bone marrow, while they could be present in plasmacytoma, formed by another tumor clone. The genetic profiling of tumor cells from different locations in the same patient (blood, bone marrow, plasmacytoma) is a promising approach to resolving this issue. The genetic heterogeneity of MM may explain cases in which it is possible to sanitize the bone marrow, and achieve immunochemical remission of the disease, but at the same time fail in treating the plasmacytoma.

Dissociation between tumor plasma cells and the stromal microenvironment of the bone marrow due to the loss of cell adhesion molecules (CD56, CD44, VLA-4) and chemokine receptors (CXCR4) on the surface of the myeloma cell are considered possible causes of the development of extramedullary lesions in MM [3,4,5]. Involvement of heparanase-1 (HPSE 1) enzyme, which destroys heparan sulfates of the extracellular matrix and stimulates angiogenesis, is assumed to promote tumor dissemination [6]. Mutations in the RAS-RAF-MAP kinase pathway genes may also trigger the spread of myeloma cells outside the bone marrow [7]. In MM patients with extramedullary plasmacytomas, compared to patients without them, high-risk cytogenetic abnormalities are significantly more often observed. Aberrations such as t(4;14) and amplification of 1q21 occur in more than 50% of patients with extramedullary lesions, and *MYC* gene rearrangements and del17p13 occur in more than 35% of patients [8,9,10]. On the contrary, in the general population of MM patients, t(4;14) is detected in 15% of cases, and the incidence of del17p13 does not exceed 10%.

Abnormalities such as del13q and del1p, characterized by the loss of extended segments of the chromosome, are well known in MM [11]. Lesions affecting shorter segments, which may result from the disturbance of homologous recombination, have been less investigated. Failure of DNA repair associated with the deficit in the homologous recombination leads to the induction of point mutations or deletions, and, as a consequence, to the progression of various malignancies [12,13]. The regions of chromosomal deletion can be determined by the loss of heterozygosity (LOH) of certain genetic loci. The loss of one of the alleles can be revealed by various molecular genetic approaches, including highly sensitive techniques that allow the determination of submicroscopic deletions and duplications (for example, comparative genomic hybridization on a microarrays). Short tandem repeats (STRs) profiling is a method suitable for LOH detection. At the same time, the method is cost- and labor-effective, less dependent of DNA sample quality, and can be used for routine screening. STRs, also known as microsatellites, are repetitive DNA sequences of 1–9 nucleotide pairs in length, make up about 3% of the human genome, are stable during the life of an individual, and, due to the high polymorphism of alleles in the population, are represented in humans mainly in a heterozygous form. Comparing the STR profiles of DNA from healthy and tumor cells could reveal LOH characteristics for certain types of malignancy.

Evaluation of plasma ctDNA in clinical practice is currently being actively investigated, both as an additional tool for predicting the course of the disease and assessing the minimal residual disease (MRD) [14]. The advantages of a liquid biopsy are obvious: the procedure is minimally invasive, does not require special training, and can be repeated many times. The use of ctDNA as a marker of MRD in solid cancers and lymphoma with no bone marrow involvement is certainly a promising direction [15,16,17]. At the same time, the value of plasma ctDNA as a marker for MRD in MM requires further clarification. In a French study published in 2018, paired plasma and bone marrow samples of 42 MM patients were compared and the MRD was evaluated. It was shown that MRD measures were the same for plasma ctDNA and bone marrow in only 49% of cases. The most frequent discrepancy observed was undetectable MRD in plasma in cases with MRD found in bone marrow [18]. The use of additional molecular targets (for example, recurrent mutations and/or copy number variations) may improve the applicability of this method for MRD detection in MM [19,20].

The aim of this study was to compare STR profiles in plasma ctDNA, CD138+ bone marrow cells, and plasmacytomas from MM patients. Additionally, we assessed *KRAS, NRAS*, and *BRAF* gene mutation profiles in the same DNA samples. It was assumed that STR profiling of DNA from plasmacytomas might reveal possible markers of extramedullary dissemination that also could be detected in plasma ctDNA. On the contrary, in patients without plasmacytomas, it was expected to see similar STR profiles in bone marrow and plasma ctDNA.

## 2. Materials and Methods

The prospective single-center study included 38 patients with symptomatic MM (16 men, 22 women) aged 35 to 84 years (median—58) newly diagnosed between 09/21/2021 and 07/21/2022. The diagnosis was established in accordance with the criteria of IMWG-2014. Plasmacytomas were detected in 66% of patients at the onset of MM (in 24 patients—bone and in 1 patient—extramedullary); in 34% of patients according to instrumental (CT/MRI) investigation plasmacytomas were not detected. Table 1 shows some clinical and laboratory parameters of patients at the onset of the disease. The majority of patients (55%) were diagnosed with stage III of the disease according to Durie–Salmon. According to the ISS classification, stages I and II were determined in an equal number of patients (29%), stage III, in 24% of patients, and in another 18% of patients, the stage was not established for technical reasons. The median hemoglobin at the onset of the disease was 105.5 g/L, and the median LDH was 162 units/L. The median share of plasma cells in the myelogram was 24%.

### 2.1. The Cytogenetic Study

A positive immunomagnetic selection of CD138+ bone marrow cells was performed using a monoclonal antibody to CD138 (Miltenyi Biotec, Bergisch Gladbach, Germany) according to the manufacturer’s protocol. The median purity of the obtained cell fraction was 27%. A FISH study of CD138+ cells was performed using DNA probes to detect translocations of 14q32/IgH, 8q14/MYC; deletions of 17p13/TP53, 13q14, 1p32; amplification of 1q21; and multiple trisomies (MetaSystems, Altlussheim, Germany). Upon detection of t(4;14), t(14;16), del17p13, amplification of 1q21, the patient was assigned to a high cytogenetic risk group.

### 2.2. The Molecular Genetic Study

For molecular genetic studies, DNA was isolated from samples of various localization (blood plasma, CD138+ bone marrow cells, buccal epithelium cells) in all patients. In addition, plasmacytoma DNA was isolated from a tumor biopsy in five patients.

### 2.3. STR Profiling

Tumor DNA STR profile was compared to the STR profile of the control DNA isolated from the buccal epithelium using multiplex STR-PCR (COrDIS Plus kit (LLC “Gordiz”, Russia), followed by fragment analysis on the on the Nanophor 05 genetic analyzer (Institute for Analytical Instrumentation Russian Academy of Science, Russia). In all patients with a change in the allele balance in the heterozygous STR loci of tumor DNA relative to the control, a relative decrease in the level of the fluorescent signal of the minor allele (in percent) was calculated (Appendix A). With 80–90% loss of the allele, the event was considered equally likely to be a deletion and a copy number neutral LOH. With a decrease in the proportion of the allele to 40–50%, the event was considered to be a duplication of the major allele in tumor cells (the ratio of alleles is 2:1 and an admixture of a small number of healthy cells). This assumption seems to be correct, since the proportion of healthy cells in the tumor biopsy is small, and in almost all patients it could be determined by focusing on loci with high rates of LOH.

### 2.4. The Mutational Status of Genes

The mutational status of *KRAS, NRAS*, and *BRAF* genes was studied in bone marrow samples of 34 patients. In 16 patients, in addition to bone marrow, mutations in these genes were evaluated in plasma ctDNA. All three localizations (plasma, bone marrow, and plasmacytoma) were studied in three patients. The mutational profile of the *KRAS* and *NRAS* genes was studied by Sanger sequencing on the Nanophor 05 genetic analyzer (Institute for Analytical Instrumentation Russian Academy of Science, Russia) and by the NGS on the MiSeq genetic analyzer (Illumina, San Diego, CA, USA). The *BRAF* V600E mutation was determined by real-time allele-specific PCR with the device CFX96 Touch (Bio-Rad, Hercules, CA, USA).

Statistical analysis was carried out using frequency analysis (using conjugacy tables and the Fisher criterion).

## 3. Results

According to FISH study of CD138+ bone marrow cells, 39.5% of patients (n = 15) were assigned to the high cytogenetic risk group, and 60.5% of patients (n = 23) to the standard risk group.

The main objective of the study was to study and compare STR profiles of tumors of various localization (plasma cDNA, CD138+ bone marrow cells, and plasmacytoma DNA). In each sample, 21 STR loci corresponding to certain chromosomes were examined. Cases with complete coincidence between plasma ctDNA and DNA isolated from CD138+ bone marrow cells were not revealed.

In 21 patients out of 38 (55%), plasma ctDNA with aberrant STR loci corresponding to LOH were detected. In the study of CD138+ bone marrow cells, loci with LOH were detected in 27 out of 38 patients (71%). Although the number of plasmacytoma cell samples studied was small, aberrant STR loci were detected in all cases analyzed (Figure 1).

Quantitative assessment of STR loci with LOH in different samples is shown in Table 2. Thus, LOH at 1–2 STR loci was detected in 26% of patients in plasma ctDNA and in 42% of patients in the DNA from CD138+ bone marrow cells. LOH at 3–4 STR loci was found in approximately an equal number of patients in plasma ctDNA and in CD138+ bone marrow cells. A large number of aberrant loci (5–8) were detected in 8% and 11% of patients in plasma ctDNA and in bone marrow, respectively.

Interestingly, the detection of a large number of aberrant STR loci in the patient’s plasma ctDNA was not a sign of the detection of similar changes in the bone marrow, and vice versa.

In cases with successful isolation of plasma ctDNA, LOH was identified in one to seven STR loci (median—three). In the DNAs isolated from CD138+ bone marrow cells aberrant STR loci were found in one to eight loci (median—two).

We analyzed the frequency of occurrence of various STR loci with LOH corresponding to a certain part of the chromosome in informative plasma ctDNA samples and bone marrow (Figure 2). Thus, it was noted that LOH is often detected at STR loci corresponding to localization on chromosomes 1 (1q42), 5 (5q23.2, 5q33.1), and 13 (13q31.1). LOH was rarely detected on chromosomes 10 (10q26.3), 18 (18q21.33), 2 (2p25.3), or Yp11.2. None of the analyzed samples of different localizations showed LOH at the STR locus corresponding to chromosome Xp22.1–22.3.

In total, 14 analyzed samples showed LOH on chromosome 1-1q42. These tumor samples belonged to 10 patients, 7 of whom were with plasmacytomas. In 4 out of 10 patients STR profiles were identical for plasma ctDNA and bone marrow.

It should be noted that LOH on chromosome 3 (3p21.31) was found only in bone marrow samples, but not in plasma ctDNA.

LOH at locus 8q24.13 was observed in samples taken from six patients, while only one of them had no plasmacytoma at the onset of MM. Similar results were observed for LOH at the 12p13.2 locus: out of seven patients, plasmacytomas were found in six.

Interestingly, LOH at the 5q33.1 locus was found in 10 samples (5 cases each for plasma ctDNA and bone marrow) belonging to 9 patients; coincidence of the marker in plasma ctDNA and bone marrow was found in one patient only.

LOH at locus 4q31.3 was noted in samples of five patients, all of them with plasmacytomas.

Most common STR aberrations in plasma ctDNA and bone marrow were evaluated in informative samples. Figure 3 shows the ratio of duplication and deletion (or copy number neutral LOH) in tumors of different localizations. As can be seen from the diagram, the most frequently detected aberrant STR markers were due to duplication.

The coincidence of aberrant STR loci in paired plasma ctDNA and bone marrow samples, depending on the presence of plasmacytomas, was evaluated (Table 3). It could be speculated that patients with plasmacytomas would have a greater variety of aberrant loci in plasma ctDNA and in the bone marrow, whereas patients without plasmacytomas would more often have similar STR loci aberrations in two tumor locations. However, no differences in the LOH frequency for different tumor locations between MM patients with and without plasmacytomas were found.

Table 4 and Figure 4 show STR loci with LOH in samples from different localizations in three MM patients. In patient No. 1, a large number of loci with LOH were detected in plasma ctDNA and bone marrow, and only two loci on chromosomes 5 and Y coincided. In the plasma ctDNA, five aberrant loci with LOH were identified that did not occur in the bone marrow, and, on the contrary, six loci with LOH on other chromosomes were identified in the bone marrow sample. This patient had multiple plasmacytomas in various bones of the skeleton. In patient No. 2 with a single bone plasmacytoma, almost complete coincidence of six STR markers in plasma ctDNA and bone marrow was observed. No plasmacytomas were detected in patient No. 3, however, the STR profiles of tumor DNA from different localizations were significantly diverse.

A considerable number of aberrant STR loci with LOH in informative tumor samples of various localizations in patients with and without plasmacytomas was analyzed; no differences were found. In plasma ctDNA from patients with plasmacytomas, one to seven aberrant loci were detected (median—2.5), and in patients without plasmacytomas, one to six aberrant loci were detected (median—3). Data for bone marrow samples also did not differ: in patients with plasmacytomas, one to eight STR aberrant loci were detected (median—two), and in patients without plasmacytomas, one to four aberrant loci were detected (median—two).

Subsequently, the entire group of patients was divided into two subgroups, depending on the number of detected loci with LOH. The first subgroup included 11 patients who had at least 4 to 8 STR loci with LOH in at least one tumor localization, and the second subgroup included 27 patients with a small number of aberrant STR loci or their absence (see Table 5). In both subgroups with high and low frequency of LOH, a comparable number of patients with plasmacytomas was noted, and these had similar LDH values and plasma cell counts. The median of the ISS stage also did not differ between the subgroups and was two. In addition, a comparable number of patients with kidney damage (stage IIIB D-S) was noted in the subgroups. There was a tendency toward lower hemoglobin values, as well as more frequent detection of high-risk cytogenetic anomalies, in the group of patients with a large number LOH loci, although the differences were not significant.

Next, the STR markers for three tumor localizations in patients with available plasmacytoma biopsies were compared (see Table 6). In most patients, plasma ctDNA was not informative. As can be seen from the table, the number of aberrant STR loci found in the DNA of plasmacytomas were higher than those in DNA isolated from bone marrow. The loci identified in the bone marrow were also found in the DNA of plasmacytomas.

*RAS* family genes mutation analysis showed that the frequency of *KRAS* gene mutation in the bone marrow substrate was 17.6% (6/34); *NRAS* gene—14.7% (5/34); and *BRAF* gene—8.8% (3/34). In general, the mutated status of *RAS* family genes was determined in 41% of patients with newly diagnosed MM.

Paired tumor samples (plasma ctDNA and CD138+ bone marrow cells) were analyzed in 16 patients with MM. In 13 patients, mutations of any of the three genes were found in the bone marrow, while in five patients, similar mutations were detected in a paired sample of tumor cDNA in plasma. Thus, in the presence of *KRAS*, *NRAS*, or *BRAF* gene mutations in the bone marrow, the frequency of their detection in plasma ctDNA was 38.5%. No cases with *KRAS*, *NRAS*, or *BRAF* gene mutation detected in the plasma and the absence of the corresponding mutation in the bone marrow were found. In three patients, the mutational status of genes was studied in three tumor localities: in the ctDNA in plasma, CD138+ bone marrow cells, and plasmacytoma substrate. Two of these had *RAS* family genes mutations in the plasmacytoma, while the genes were not mutated in plasma ctDNA and in the bone marrow substrate. *KRAS* gene mutation was detected in seven patients (in six bone marrow samples and one plasmacytoma sample), and *NRAS* gene mutation was detected in six patients (in five bone marrow samples and one plasmacytoma sample). It should be noted that only 14% of patients with mutated *KRAS* genes were assigned to high cytogenetic risk, whereas in those with *NRAS* gene mutation, 67% were at high cytogenetic risk.

## 4. Discussion

Tumor cell dissemination in various neoplasms, including MM, is associated with genome instability in general, and LOH particularly. Various genetic defects, including an increase in the number LOH loci, accumulate during evolution of tumor clones. Over time, various tumor clones with altered ability for metastasis, invasion, and chemoresistance may emerge. Those overcoming host immunological surveillance are positively selected. In MM, the heterogeneity of genetic aberrations in tumor cells localized in anatomically diverse locations has been previously reported [21].

Previously, we studied STR profiles of tumor cells in acute lymphoblastic leukemia and aggressive lymphomas [22,23]. Aberrant STR profiles were found in 20% of patients with acute leukemia and 50% of patients with lymphomas. Here, we report data concerning STR profiles of anatomically diverse tumor loci in patients with newly diagnosed symptomatic MM. Aberrant STR profiles are noted for most of the patients; moreover, they differ between tumor loci. In more than half of patients (55%), we detected markers with LOH in plasma ctDNA. In 71% of patients, LOH was found in CD138+ bone marrow cells. Despite the small number of analyzed plasmacytomas biopsies, it is noteworthy that aberrant STR loci with LOH were detected in all samples.

An average of two–three loci with LOH were detected in a tumor sample. It is important that the detection of LOH in seven–eight STR loci in plasma ctDNA did not mean that the bone marrow of this patient would have an identical picture. Of all samples analyzed, in no case was a complete coincidence between STR profiles of plasma ctDNA and CD138+ bone marrow cells or plasmacytoma observed. This indicates the uniqueness of MM as a genetically heterogeneous tumor.

Most often, LOH was detected at the STR locus corresponding to chromosome 1 (1q42), which is logical considering that more than 30% of the “high risk MM” genes are located on chromosome 1 [24].

It was also observed that samples with LOH on chromosomes 8 (8q24.13), 12 (12p13.2), and 4 (4q31.3) mainly belong to patients with plasmacytomas detected at the onset of disease. The study of extended cohorts of patients may reveal possible LOH markers associated with plasmacytoma development.

Initially, we assumed that the STR profile of the plasma ctDNA and bone marrow would differ in MM patients depending on the presence of plasmacytomas. Considering that the tumor cells acquire the ability to function outside the bone marrow during the formation of a plasmacytoma, it was expected to see a greater variety of aberrant loci in plasma ctDNA and in bone marrow in patients with plasmacytomas. This hypothesis was not confirmed—no difference in distributions of STR loci with LOH between MM patients with or without plasmacytomas was detected. This indicates the genetic diversity of tumor clones in MM, regardless of the presence of radiologically confirmed plasmacytomas. Therefore, we conclude that risk stratification based on molecular tests performed solely on bone marrow samples may not be sufficient for all MM patients, including those without plasmacytomas. Due to genetic heterogeneity of MM tumor cells from various lesions, the high diagnostic value of liquid biopsy approaches becomes obvious. The range of mandatory diagnostic measures for MM does not include a biopsy of a plasmacytoma in cases with proven bone marrow involvement. A patient may have several plasmacytomas of different localizations at the same time, and it may be unsafe and impractical to perform multiple biopsies. The study of plasma ctDNA may replace invasive techniques in these cases. One can expect that the study of plasma ctDNA will be used not only for scientific purposes, but also may become an indispensable tool in determining high-risk factors in routine MM clinical practice. Additionally, it is proposed to use plasma ctDNA profiling for the prediction of MM recurrence. Binod Dhakal et al. retrospectively analyzed 80 blood samples taken at different time points in 28 patients with MM after auto-HSCT. The researchers showed that plasma ctDNA was detected in 70.8% of patients before transplantation and in 53.6% of patients after HSCT. Out of 15 patients who were diagnosed with plasma ctDNA after HSCT, 14 developed a relapse at follow-up [25]. However, further prospective studies on extended patient cohorts are still required to determine the value of plasma ctDNA as a marker of early relapse after auto-HSCT.

We report frequent deletions at STR loci on chromosomes 8 (8q24.13), 12 (12p13.2), 13(13q31.1), 16 (16q24.1), and 6 (6q14) in MM samples. These data are consistent with the study by Brian A. Walker et al., who noted a high incidence of deletions at chromosomes 12, 13, 16, and 6. The authors also note frequent deletions at loci 1p, 8p, 14q, 17p, 20, and 22 in MM patients [11].

Researchers from the University of Arkansas studied LOH in tumor samples taken from 406 MM patients. The expression profile of 405 genes associated with the pathogenesis of various malignancies was also studied using new generation sequencing. The group of patients was heterogeneous and included patients with MGUS, smoldering MM, newly diagnosed symptomatic MM, and recurrent/refractory MM. The researchers analyzed LOH associated with a deficiency of homologous recombination, suggesting that in the presence of this phenomenon, PARP inhibitors (inhibitors of the enzyme poly ADP ribosopolymerase) could potentially be used in the treatment of MM. It was shown that with the transformation of the disease from MGUS to recurrent/refractory MM, the frequency of LOH detection increases. An association has also been demonstrated between the frequency of LOH and the gene expression profile characteristic for high-risk MM. In patients with a high LOH frequency, lower indicators of survival were noted compared with patients having low LOH frequency. The authors conclude that mutations in key homologous recombination genes may explain some, but not all, cases of high LOH frequency, and that further association studies are still required [26].

We analyzed subgroups of patients in regard to the number of detected aberrant loci with microsatellite instability in plasma ctDNA and bone marrow. We report a tendency toward lower hemoglobin values, as well as more frequent detection of high-risk cytogenetic anomalies in the group of patients with a large number (4–8) LOH loci compared with the subgroup of patients with a smaller number (0–3) of aberrant markers. These results are comparable with the data obtained by other scientific groups [26].

According to published data, mutations of *RAS* family genes (*KRAS*, *NRAS*, *BRAF*) occur in less than 10% in MGUS, in 50% of symptomatic MM, and in more than 70% of cases of plasma-cell leukemia, which indicates activation of this pathway in the progression of the disease [7,27]. We report that 41% of patients with newly diagnosed MM have mutations in the *RAS* family genes. *KRAS*, *NRAS*, or *BRAF* gene mutations were detected in the bone marrow substrate for 38.5% of patients; similar data were obtained for plasma ctDNA. Despite the small number of plasmacytoma samples analyzed, in two out of three cases the mutation status of the genes was different from those in bone marrow. *NRAS* or *KRAS* genes were mutated in plasmacytomas, while in the bone marrow or plasma ctDNA the genes were not affected. The difference found may indicate the presence of different clones in different tumor locations.

Despite the fact that activating mutations in the oncogenes of the *RAS* family are quite often detected in MM, the data on their role in pathogenesis are contradictory. Although mutations in *RAS* genes are associated with a more aggressive phenotype and lower survival indicators, they are rarely found in combination with other high-risk MM factors such as t(4;14) and deletion 17p13. At the same time, the association between the mutated status of *RAS* family genes and the presence of t(11;14) may exist. In these cases, a combination with the *RAS* family gene mutation is associated with decreased survival, while t(11;14) alone refers to standard risk MM [28]. Despite the small number of observations, high-risk cytogenetic aberrations were more frequent in patients with *NRAS* gene mutations compared to the those with *KRAS* gene mutations (67% vs. 14%).

## 5. Conclusions

The study of the MM genetic landscape is a promising area of fundamental hematology. The study of genetic features of tumor plasma cells from different localizations in the same patient also may open up new opportunities in clinical practice. Risk stratification based on molecular tests performed solely on bone marrow samples may not be sufficient for all MM patients, including those without plasmacytomas. Due to genetic heterogeneity of MM tumor cells from various lesions, the high diagnostic value of liquid biopsy approaches becomes obvious. It seems important to analyze LOH profiles in association with first line therapy response, high-dose chemotherapy/auto-HSCT outcome, and survival indicators. This prospective research is currently in progress. We also plan to extend the patient cohort to detect significant indicators regarding association of *RAS* family gene mutations with other risk factors in MM.

## Figures and Tables

**Figure 1 genes-14-00351-f001:**
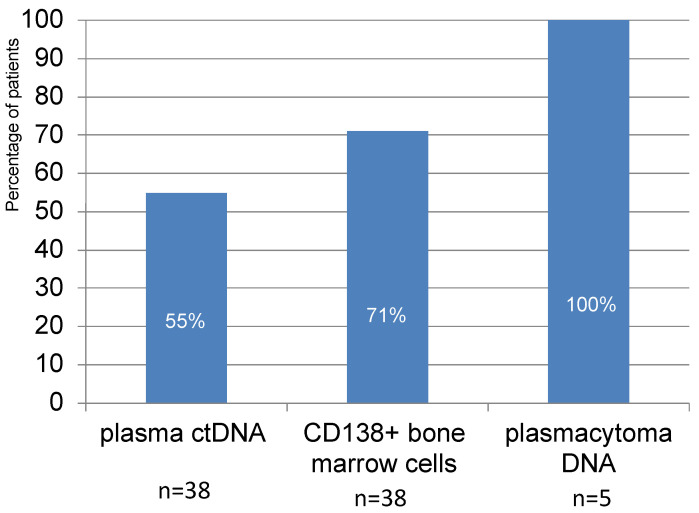
Frequency of aberrant STR loci occurrence in the samples from different localizations in MM patients.

**Figure 2 genes-14-00351-f002:**
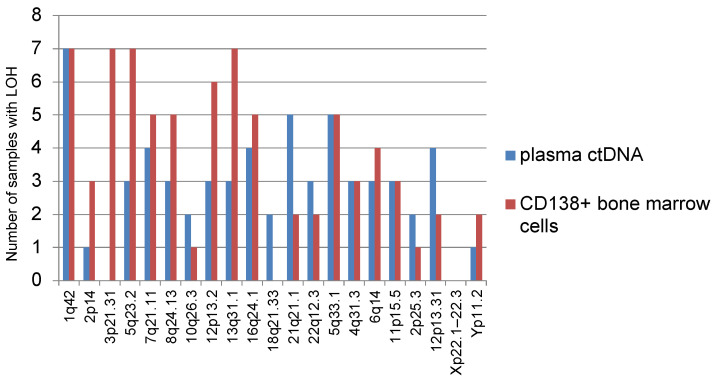
Chromosomal localization of LOH in plasma ctDNA and bone marrow samples.

**Figure 3 genes-14-00351-f003:**
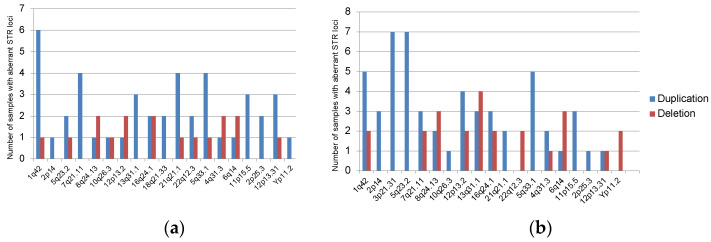
Occurrence of duplication and deletion (or copy number neutral LOH) in plasma ctDNA (**a**) and in CD138+ bone marrow cells (**b**).

**Figure 4 genes-14-00351-f004:**
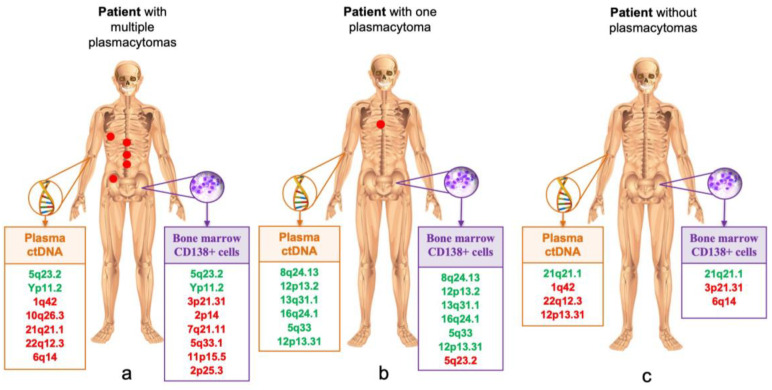
Comparison of STR profiles in plasma ctDNA and CD138+ bone marrow cells in MM patients. Plasmacytomas are schematically depicted with a red circle. Matching markers are highlighted in green, different markers are highlighted in red. (**a**) a large number of different markers were detected; (**b**) almost identical STR profile; (**c**) different STR profile. *STR*—*short tandem repeats; ctDNA—circulating tumor DNA; MM*—*multiple myeloma*.

**Table 1 genes-14-00351-t001:** Characteristics of MM patients.

Parameters	Patients with MM n = 38
Age, years, median and range	58 (35–84)
Males/females	16/22
Type of secretion	
G	25 (66%)
A	7 (18%)
BJ	5 (13%)
D	1 (3%)
Type of FLC	
κ	26(68%)
λ	12 (32%)
D-S stage	
IA	3 (8%)
IB	1 (3%)
IIA	4 (10%)
IIIA	21 (55%)
IIIB	9 (24%)
ISS stage	
I	11 (29%)
II	11 (29%)
III	9 (24%)
Not available	7 (18%)
Hemoglobin (g/L), median and range	105.5 (66–129)
LDH (U/L), median and range	162 (73–694)
% plasma cells in bone marrow aspiration, median and range	24 (3–92)
Purity of CD138+ cells, percentage, median and range	27 (8–92.4)
Plasmacytomas	
Yes	25(66%)
No	13 (34%)

**Table 2 genes-14-00351-t002:** Quantitative assessment of STR loci with LOH in plasma ctDNA and in CD138+ bone marrow cells of MM patients.

The Number of Aberrant STR Loci	Plasma ctDNA n = 38	CD138+ Bone Marrow Cells n = 38
0	17 (45%)	11 (29%)
1–2	10 (26%)	16 (42%)
3–4	8 (21%)	7 (18%)
5–8	3 (8%)	4 (11%)

**Table 3 genes-14-00351-t003:** LOH frequency in paired plasma ctDNA and bone marrow samples, depending on the presence of plasmacytomas.

Parameter	MM with plasmacytomas n = 25	MM without plasmacytomas n = 13	*p*
The coincidence of one or more STR loci in paired plasma ctDNA and bone marrow samples	8 (32%)	5 (38%)	0.73

**Table 4 genes-14-00351-t004:** Comparison of STR profiles in plasma ctDNA and CD138+ bone marrow cells in MM patients.

Patient Number	Chromosomal Localization of STR Loci	LOH		Localization of Plasmacytomas
		Plasma ctDNA	CD138+ Bone Marrow Cells	
1	5q23.2	+	+	10th rib, Th12, L3, L4, ilium
	Yp11.2	+	+	
	1q42	+	-	
	10q26.3	+	-	
	21q21.1	+	-	
	22q12.3	+	-	
	6q14	+	-	
	3p21.31	-	+	
	2p14	-	+	
	7q21.11	-	+	
	5q33.1	-	+	
	11p15.5	-	+	
	2p25.3	-	+	
2	5q23.2	-	+	Th4
	8q24.13	+	+	
	12p13.2	+	+	
	13q31.1	+	+	
	16q24.1	+	+	
	5q33	+	+	
	12p13.31	+	+	
3	21q21.1	+	+	No plasmacytomas
	1q42	+	-	
	22q12.3	+	-	
	12p13.31	+	-	
	3p21.31	-	+	
	6q14	-	+	

**Table 5 genes-14-00351-t005:** Clinical and laboratory parameters in patients with different LOH frequencies.

Parameters	Patients with 4–8 Loci with LOH (n = 11)	Patients with 0–3 Loci with LOH (n = 27)
Plasmacytomas		
yes	7 (63.6%)	18(66.4%)
no	4 (36.4%)	9 (33.3%)
% plasma cells in bone marrow aspiration, median and range	23 (6–68)	27 (3–92)
LDH (U/L), median and range	160 (111–262)	166 (73–694)
D-S stage		
IIIB	3 (27%)	6 (22%)
Median of ISS stage	II	II
Hemoglobin (g/L), median and range	95 (66–124)	106 (71–172)
High-risk cytogenetics	5 (45.5%)	10 (37%)

**Table 6 genes-14-00351-t006:** Comparison of aberrant STR loci in tumors of different localizations in MM patients.

Patient Number	Detected STR Locus (Number and Chromosomal Localization)		
	Plasma ctDNA	CD138+ Bone Marrow Cells	DNA of Plasmacytomas
1	1 1q42	2 1q42 10q26.3	3 1q42 10q26.3 22q12.3
2	0	2 5q23.2 5q33.1	4 1q42 5q23.2 5q33.1 11p15.5
3	0	2 8q24.3 13q31.1	2 8q24.3 13q31.1
4	0	0	7 3p21.31, 5q23.2, 16q24.1, 18q21.33, 5q33.1, 6q14, 11p15.5
5	0	2 12p13.2 4q31.3	1 12p13.2

## Data Availability

The following Appendix A can be downloaded at Appendix A.

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
