# Peer review of "Loss of Heterozygosity in the Circulating Tumor DNA and CD138+ Bone Marrow Cells in Multiple Myeloma"

_genes, 2023, doi:10.3390/genes14020351_

Round 1

Reviewer 1 Report

The manuscript is well written. The STR profiling was used as the main tool to detect Loss of heterozygosity (LOH) in the circulating tumor DNA and 2 CD138+ bone marrow cells in multiple myeloma. Multiple genes at multiple loci have been considered for the comparative analysis.

The study is based on single institution and ct DNA analysis of all the patient samples were not available to correlate the analysis. So in addition to STR profiling more experimental analysis could have been done to support the result and robustness of the concluding statement.

Method section may be clearly written under suitable subheadings. The catalogue numbers or model numbers of the machines used for  the experiment should be mentioned for the clarity. 

More diagramatic representation will be appreciated. 

Author Response

Dear Reviewer, thanks for your work. Your valuable comments will help us in the future, as the study continues; we will try to conduct additional experimental analysis.
I added subheadings to the Method section and indicated the model numbers of the machines.
In the results section, I have added a diagram (Figure 4) to helps understand Table 4
Thank you very much

Reviewer 2 Report

In this article authors have studied differences in STR profiles of different MM lesions and suggest that risk stratification based on bone marrow biopsy alone might not be sufficient.

Overall article has good scientific soundness.

I suggest removing figure 2 as data in figure 2 and table 2 are same. 

Author Response

Dear Reviewer, thanks for your work.
I deleted fig. 2
In addition, I added subheadings to materials and methods.
In the results section, I have added a diagram (Figure 4) to helps understand Table 4
Thank you very much